# Genetic Diversity and Distribution of Italian Cave Crickets (*Dolichopoda*): Toward a Better Understanding of Lineage Structure

**DOI:** 10.3390/ani15162429

**Published:** 2025-08-19

**Authors:** Matteo Garzia, Emanuele Berrilli, Enrico Lunghi, Luca Coppari, Nathan Delcour, Daniele Salvi

**Affiliations:** Department of Health, Life & Environmental Sciences, University of L’Aquila, Via Vetoio snc, Coppito, 67100 L’Aquila, Italy; matteo.garzia@univaq.it (M.G.); emanuele.berrilli@univaq.it (E.B.); enrico.lunghi@univaq.it (E.L.); luca.coppari@graduate.univaq.it (L.C.); nathandelcour.bio@gmail.com (N.D.)

**Keywords:** cox1, 16S, phylogeography, molecular systematics, Orthoptera, Appennine diversity, Dolichopodinae

## Abstract

Cave crickets of the genus *Dolichopoda* are key components of Mediterranean cave ecosystems, especially in Italy. Despite their importance, much of their diversity remains unexplored. In this study, we analyzed mitochondrial DNA from 18 populations across the Apennines and identified six known species, new genetic lineages within *D. geniculata*, and several range extensions. We also report the first evidence of co-occurrence between two *Dolichopoda* species. These results highlight the need for further research to refine taxonomy and guide conservation efforts.

## 1. Introduction

The cave crickets (Orthoptera: Rhaphidophoridae) are considered model organisms for biogeography because their evolutionary patterns are strongly shaped by vicariant events, making them ideal for studying historical biogeographic processes across both broad and fine spatial scales [1,2]. According to Allegrucci et al. [1], the split between the two Mediterranean subfamilies Dolichopodinae and Troglophilinae is estimated to have occurred approximately 40 million years ago (Ma). The genus *Dolichopoda* (Dolichopodinae) currently comprises 67 recognized species, distributed from Southwestern Europe across the Anatolian region and extending as far as Iran [3]. The complex palaeogeographic history of the Mediterranean region has likely played a pivotal role in shaping the colonization dynamics and speciation patterns within the genus *Dolichopoda.* This genus is characterized by species that occupy a variety of microhabitats, including forest soil crevices, catacombs, Etruscan tombs, anthropogenic subterranean structures, natural caves, and extensive hypogean karst systems—ranging from quasi-epigean to fully hypogean ecological conditions [4,5]. The variety of microhabitats also reflects the variety of diet (e.g., dead vegetation, small arthropods, and bat guano), and this demonstrates the key role of cave crickets in the food web of several different semi-subterranean and hypogean ecosystems, transferring energy from detrital resources to higher trophic levels [6]. The biogeographic history of the genus was widely studied, and an eastern origin of *Dolichopoda* species has been estimated [7]. The sylvicolous ancestors of *Dolichopoda* likely used caves as refuges during past climatic fluctuations during the Pliocene and Pleistocene, and speciation within each phylogeographic unit may have occurred primarily driven by vicariant events. Furthermore, the current species distribution of the genus *Dolichopoda* is best explained by a combination of vicariance and dispersal, with key biogeographical processes occurring in epigean populations prior to cave colonization [7].

Italy hosts the highest diversity of West Mediterranean *Dolichopoda* species. In particular, nine species of *Dolichopoda* occur in Italy, from the Maritime Alps to the southern tip of the Italian peninsula: *D. schiavazzii*, *D*. *azami, D. aegilion*, *D. apollinea*, *D. baccettii*, *D. laetitiae*, *D. geniculata*, *D. capreensis*, and *D. palpata* [2,8,9]. Most of the Italian species present an allopatric distribution with several cases of insular endemic species and subspecies present in a localized area of the Apennines [8,10,11]. Furthermore, the study of Di Russo et al. [2] discovered a new species in southern Italy and presented the most updated distributions of *Dolichopoda* species in the southernmost Italian region. This study demonstrated that the Italian *Dolichopoda* diversity is still understudied and that deepening the sampling coverage allows new discoveries on the genetics, distribution, and taxonomy of the group. In particular, the northern and central areas of the Apennines, hosts of most of the *Dolichopoda* diversity, remained unstudied for almost 20 years [8].

The Apennine region is one of the most important European areas for biodiversity conservation due to its past role as a glacial refugia [12,13,14,15]. However, the limited attention given to invertebrates has left their distribution, genetic diversity, and, consequently, their conservation status largely unexplored [16]. Nevertheless, invertebrates are known to be useful predictors of conservation priorities, which can also benefit vertebrate conservation [17]. Thus, improving our knowledge about cave invertebrates will help to address new conservation efforts on cave biodiversity within the Apennines.

In this study, we sequenced *Dolichopoda* specimens from different localities along the Apennine range to better characterize the distribution of species present in Italy. We used two mitochondrial markers, 16S and *cox1*, which are particularly useful for assessing the genetic diversity of cave-dwelling species for two main reasons: (i) they evolve relatively rapidly, providing high resolution for detecting phylogenetic relationships and intraspecific diversity; and (ii) they are widely used in previous studies, allowing for direct comparisons and integration with the existing data on the Italian cave fauna. This will enhance our understanding of the genetic diversity and geographic distribution of the main evolutionary lineages, providing valuable information for future evolutionary and taxonomic studies.

## 2. Material and Methods

### 2.1. Sampling

Our study included 45 specimens of *Dolichopoda* spp., representing 18 populations. Sampling of the populations was carried out in Corsica during spring 2016 and in Italy throughout the Apennine range during spring and summer 2024 (Table 1). The samples were collected, and one leg from each specimen was removed and preserved in 95% ethanol for subsequent genetic analysis.

### 2.2. DNA Extraction, Amplification, and Sequencing

Total genomic DNA was extracted from a leg fragment with a length of 2 mm using a standard high-salt protocol [18]. The details of the DNA extraction are as follows: Step 1—overnight lysis at 56 °C of the tissues using 15 μL of Proteinase K 20 mg/μL (Meridian, Bioscience) in 450 μL of lysis buffer (10 mM TRIS-HCl, 2 mM EDTA, and 1% SDS); Step 2—fast spin for 10 s, select the clean surfactant, put it into a new tube, and discard the chitinous exoskeleton; Step 3—add 160 μL of NaCl solution 5 M at 4 °C, and mix well; Step 4—centrifuge at 4 °C for 15 min, select the clean surfactant, put it into a new tube, and discard the previous tube with the residue of salt on the bottom; Step 5—add 650 μL of cold isopropanol, mix carefully, and keep at 4 °C for up to two hours; Step 6—centrifuge at 4 °C for 30 min; Step 7—discard surfactant, and keep the tube with the pellet on the bottom; Step 8—add 1 mL of 70% ethanol; Step 9—centrifuge at room temperature for 15 min; Step 10—discard surfactant, and keep the tube with the pellet on the bottom; Step 11—evaporate the remaining ethanol into a thermoblock at 37 °C; Step 12—add 80 μL of MilliQ water, and elute the pellet. Amplification was carried out in a total volume of 25 μL, with 17.3 μL of MilliQ water, 3 μL of 10× NH4 reaction buffer, 1.5 μL of 50 mM MgCl_2_ solution, 0.5 μL of 100 mM dNTP, 0.5 μL of each primer (10 mM), 0.5 μL of BSA, 0.2 μL of BIOTAQ™ (Meridian, Bioscience, London, UK), and 1 μL (~40 ng) of DNA template. We amplified the mitochondrial cytochrome c oxidase I gene using two pairs of primers (*cox1*; primer pairs: LCO1490F/HCO2198R [19] and UEA5/UEA8 [20]) and 16S ribosomal DNA (16S rDNA; primer pairs: 16Sa/16Sb [21]), following the PCR conditions reported in Salvi et al. [22]. Successful amplification was determined by gel electrophoresis, and the PCR products were purified and sequenced based on the Sanger method by an external service (Genewitz, Leipzig, Germany). The obtained chromatograms of each sequence were manually edited and assembled into a consensus sequence using Geneious Prime 2021 (Biomatters Ltd., Auckland, New Zealand). Consensus sequences were deposited in the GenBank database (GenBank accession:PX122103-PX122152; PX124280-PX124329). 

The phylogenetic analyses were carried out in two steps. Firstly, to determine the species-level identification of the collected samples, *cox1* sequences were aligned with homologous sequences of *Dolichopoda* obtained from previous studies and available on GenBank (Appendix A). Once it was established that our samples belonged to species already known from Italy, we used both mitochondrial markers (*cox1* and 16S) to assess the phylogenetic relationships within the identified species group.

For the first step, we built a non-redundant database comprising all available *Dolichopoda* cytochrome gene sequences from the public repositories of GenBank, following the procedure described by Salvi et al. [22]. Since *cox1* was amplified in two separate fragments, we first used the “Map to Reference” option in Geneious 2022.2.2 to align both fragments against the complete mitochondrial genome of *Rhaphidophora duxiu* (GenBank accession: PP953500), allowing us to generate a single contiguous alignment. After mapping, we selected and extracted the region of the gene located between the primers LCO1490 and UEA8. A multiple sequence alignment was then performed using MAFFT v.7 with the FFT-NS-1 progressive method algorithm and default parameters [23]. The final *cox1* dataset used for the analyses comprised 300 sequences representing 42 *Dolichopoda* species.

For the second step, we separately aligned sequences of both the *cox1* and 16S genes with MAFFT v7.450 using the G-INS-I progressive method algorithm [23]. The total lengths of the *cox1* and 16S aligned datasets are 1165 bp and 484 bp, respectively. We first used the *cox1* sequences to infer a Neighbor-Joining (NJ) under the p-distance model and pairwise deletion with MEGA11 [24]. Then, concatenated sequence alignments (*cox1* + 16S) from Italian and French sequences comprising our sequences in the NJ were used to infer a Maximum Likelihood (ML) tree in IQ-TREE 1.6.12 [25] using the W-IQ-TREE webserver [26] with *Troglophilus cavicola* as the outgroup ([2,25,27,28], GenBank accession: AY793564 and AY793624 [8]). The best substitution models of each partition in our concatenated matrix of all genetic markers were determined by the ModelFinder module, including flexible rate heterogeneity across sites models [29], based on the Bayesian Information Criterion (BIC). We used the Edge Linked partition model to allow each partition to have its own evolutionary rate. Branch support was assessed by 1000 replicates of ultrafast bootstrapping (uBS) [30,31]. FigTree v1.3.1 [32] was used to depict the trees. The support thresholds for considering a topology as robust were set to >95 for ultrafast bootstrap values (uBS) in the Maximum Likelihood analysis and >0.95 for posterior probabilities (PP) in the Bayesian analysis.

## 3. Result

The phylogenetic analyses based on NJ trees of 301 *cox1* sequences allowed us to assign the sequenced specimens collected in this study to six previously recognized Italian species: *D. laetitiae*, *D. geniculata*, *D. bormansi*, *D. schiavazzii*, *D. calabra*, and *D. palpata* (Figure 1A,B).

The Maximum Likelihood analysis, based on 173 concatenated sequences of 16S and *cox1*, focusing on species with predominantly Italian distributions, confirmed the same taxonomic assignment for these specimens to six species, each represented by a well-supported clade (ultrafast Bootstrap Support, uBS > 90), with the exception of *D. geniculata*. This is represented by four distinct clades—two previously recognized as *D. geniculata* and *D. geniculata geniculata* and two additional clades corresponding to newly identified evolutionary lineages, here referred to as *D. geniculata new lineage 1* (uBS = 100) and *D. geniculata new lineage 2* (uBS = 100).

Regarding the geographic distribution of species and lineages, *D. bormansi* was found in locality 1. *D. calabra* occurred in localities 44 and 45, with 1 and 2 individuals, respectively. *D. geniculata* was present in localities 11 and 36, with 1 and 2 individuals, respectively. The nominal lineage *D. geniculata geniculata* was found in locality 40, represented by a single individual. *D. geniculata new lineage 1* was recorded in localities 11, 12, 14, 39, and 40, with 2, 4, 3, 2, and 2 individuals, respectively. *D. geniculata new lineage 2* was found exclusively in locality 46, where 2 individuals were collected. The species *D. laetitiae* was found in localities 2, 4, 17, 21, and 41, with 3 individuals at each site. *D. palpata* was recorded in locality 45, represented by 1 individual. Therefore, in site 45, *D. palpata* and *D. calabra* are syntopic. Finally, *D. schiavazzii* was identified in locality 8, with a total of 3 individuals.

Our results also expand the known distribution ranges for several species (Figure 2). Individuals of *D. laetitiae* from localities 2, 4, and 41 were found approximately 40 km west of its previously known range, in an area historically attributed to *D. schiavazzii*. Similarly, *D. geniculata* was detected in localities 11, 12, 14, and 39, extending its known range northward into the previously documented distribution area of *D. laetitiae*. Lastly, *D. calabra* was recorded in locality 45, approximately 115 km south of its previously known southern range limit.

## 4. Discussion

Understanding the genetic diversity and distributional patterns of invertebrate taxa is essential for reconstructing their evolutionary histories, identifying cryptic diversity, and informing conservation priorities. Despite the ecological importance and endemic richness of the Italian invertebrate fauna, many groups remain poorly characterized from both genetic and biogeographic perspectives [15,17,22,34]. This is particularly true for cave-dwelling taxa, such as *Dolichopoda*, which often exhibit restricted distributions and high levels of local endemism yet are underrepresented in genetic databases [9]. By incorporating newly sampled populations across the Apennine range, this study contributes genetic and geographic data that help refine species boundaries, detect potential unrecognized lineages, and update current distributional ranges. These efforts are fundamental not only for a more accurate taxonomic framework but also for understanding the processes that have shaped the diversification and persistence of subterranean faunas in complex landscapes such as the Italian peninsula.

Our phylogenetic and geographic data provide new insights into the diversity and biogeographic structure of *Dolichopoda* species along the Apennine range. The newly generated sequences confirm the presence of six previously recognized species (*D. laetitiae*, *D. geniculata*, *D. bormansi*, *D. schiavazzii*, *D. calabra*, and *D. palpata*) but also reveal the existence of novel, deeply divergent mitochondrial lineages within the *D. laetitiae*/*D. geniculata* clade and within southern taxa (*D. calabra* and *D. palpata*).

Our phylogenetic and distributional analyses provide new insights into the taxonomy and evolutionary relationships within the *Dolichopoda geniculata*–*laetitiae* clade, reinforcing previous hypotheses about its complex nature. The inclusion of newly sampled individuals from the central Apennines allowed us to further characterize the genetic structure of this group, supporting earlier findings by Martinsen et al. [35] that documented elevated intraspecific divergence in *D. geniculata* compared to other congeners. Importantly, our ML analyses recovered a strongly supported monophyletic group comprising both *D. geniculata* and *D. laetitiae*, with two newly identified and geographically structured lineages nested within this clade. Despite the current taxonomic distinction between the two species, our results, as well as those from previous studies [9,27,35], highlight the lack of clear molecular, ecological, or morphological differentiation between them.

The geography of the *geniculata*–*laetitiae* clade spans heterogeneous environments and is shaped by substantial topographic barriers, including mountain ridges, deep valleys, and isolated karst systems that characterize the central Apennine. However, genetic continuity across these regions suggests that isolation has not led to full speciation. Both taxa inhabit ecologically overlapping subterranean environments (karst caves and riparian hypogean habitats) without evidence of niche partitioning or habitat-based divergence [35]. Moreover, neither the original species descriptions nor subsequent morphological evaluations provide robust diagnostic characters separating *D. geniculata* from *D. laetitiae* [9,27,35]. This, therefore, suggests a scenario of lineage diversification within a single evolutionary entity rather than the presence of two distinct evolutionary and taxonomic units. We, thus, consider it important to further investigate the relationships within this species complex using additional molecular, morphological, and ecological evidence, as the current taxonomy may either overestimate or underestimate the actual species diversity present.

In addition to these taxonomic considerations, our expanded sampling revealed new and important aspects of the spatial distribution of these taxa. We document notable range extensions for several species, including a westward expansion of *D. laetitiae* into regions previously attributed to *D. schiavazzii* and a northward occurrence of *D. geniculata* within areas traditionally occupied by *D. laetitiae*. Similarly, *D. calabra* was recorded farther south than previously known, while *D. geniculata* was also found in novel, northern localities, further confirming its occurrence in more northern regions [33]. Particularly noteworthy is the first evidence of sympatry between *D. calabra* and *D. palpata* in locality 45, with genetically distinct lineages found in syntopy. This contact zone raises important questions about reproductive isolation (i.e., pre-/post-zygotic barriers), historical biogeography, and potential ecological interactions between these southern species, previously considered allopatric.

These findings collectively highlight the importance of broad-scale, fine-resolution sampling across the Italian Apennine, especially in underexplored environments such as caves. Subterranean environments often function as both long-term refugia and historical corridors, shaping patterns of persistence and diversification across geological timescales [7]. However, the current sampling coverage across the Apennines remains patchy, likely underestimating true diversity, range limits, and contact zones. Expanding molecular, morphological, and ecological surveys in these habitats is, therefore, critical not only to refine species boundaries and taxonomic frameworks but also to inform conservation strategies for these endemic and habitat-specialized invertebrates.

## 5. Conclusions

Our study reveals previously unrecognized genetic diversity within Italian *Dolichopoda*, particularly in the *D. geniculata* clade, and documents expanded species distributions and new contact zones. These results challenge current taxonomic boundaries and suggest a more intricate evolutionary history than previously assumed.

Continued molecular and geographic surveys in cave habitats are crucial to resolving taxonomic uncertainties and ensuring accurate assessments of biodiversity for conservation planning.

## Figures and Tables

**Figure 1 animals-15-02429-f001:**
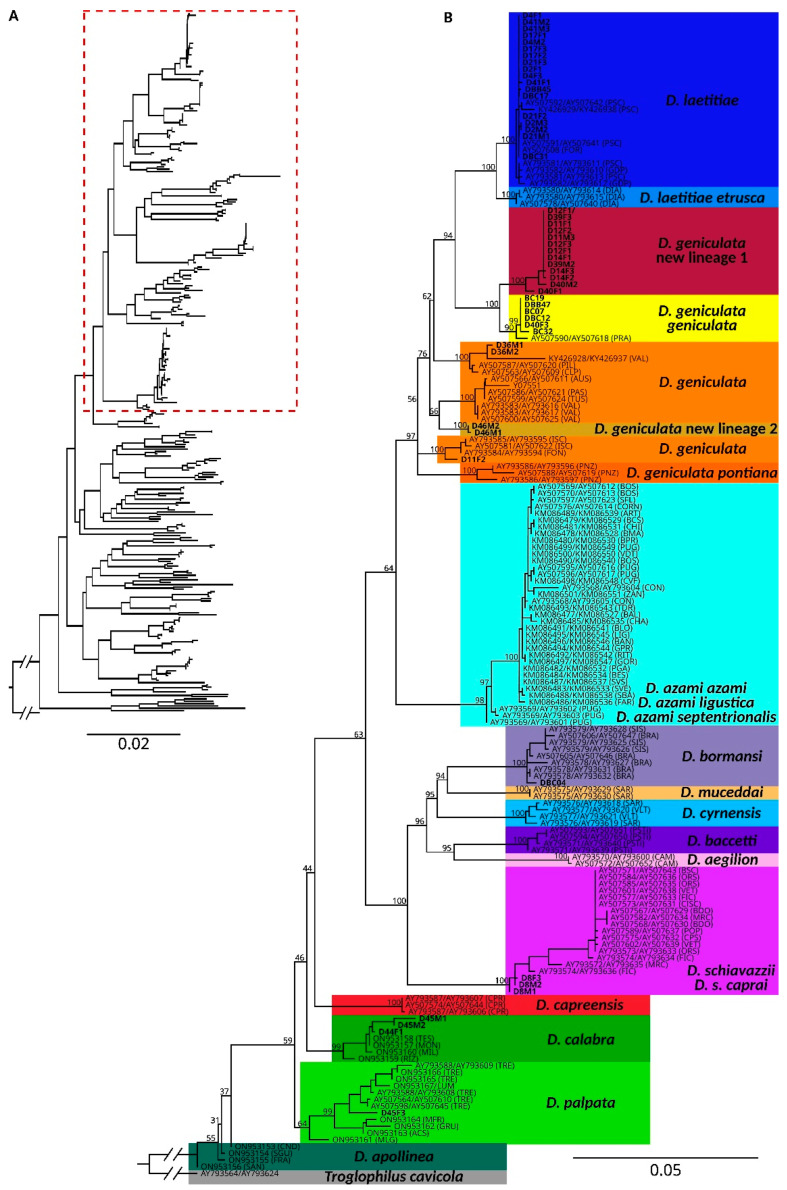
(**A**) Neighbor-Joining tree based on all available *cox1* sequences of *Dolichopoda* from GenBank. The dashed red rectangle refers to the subtree shown in section B. (**B**) Maximum Likelihood tree inferred from the concatenated *cox1* and 16S datasets, illustrating the phylogenetic relationships among the main clades, with a focus on lineages occurring in Italy. Our samples are underlined in bold. The colors represent distinct species as reported in previous studies and distinct lineages as recovered in this study.

**Figure 2 animals-15-02429-f002:**
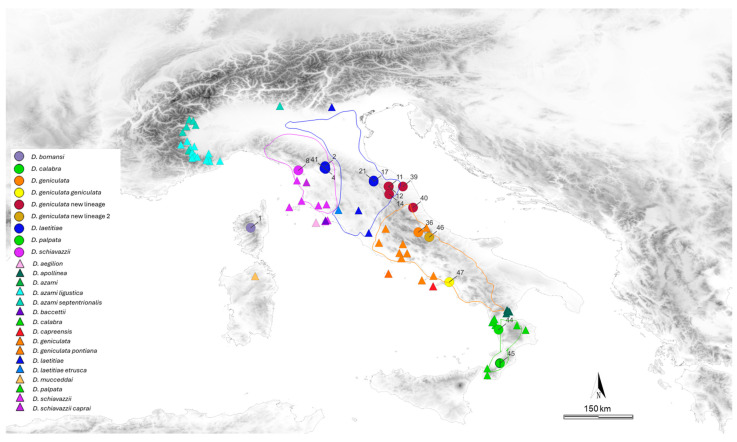
Distribution of the Italian and Corsican species within the *Dolichopoda* genus. The circles indicate newly sequenced specimens from this study; the triangles indicate sequenced specimens retrieved from GenBank. The colors follow the clades identified in the ML tree (Figure 1B); the outlines delimit the distribution of the lineages according to Allegrucci et al. [8] and Meier et al. [33].

**Table 1 animals-15-02429-t001:** Geographic information about sequenced specimens in this study. The “Locality IDs” are also reported in the map (see Results). The number of specimens in each locality is indicated under the column “N”. The column “Lat” indicates latitudes, and “Lon” indicates longitudes.

Locality ID	Locality	Species	N	Lat (°)	Lon (°)
1	Restonica	*D. bormansi*	1	42.22	9.05
2	Tasso	*D. laetitiae*	3	43.98	11.16
4	Forra	*D. laetitiae*	3	43.92	11.14
8	Maggiano	*D. schiavazzii*	3	43.86	10.4
11	Solstizio	*D. geniculata*	3	43.4	12.97
12	Grotta Leonardo	*D. geniculata*	4	43.4	12.96
14	Vurgacci	*D. geniculata*	3	43.18	12.98
17	Grotta Nottole	*D. laetitiae*	3	43.54	12.54
21	Grotta Borghetto	*D. laetitiae*	3	43.57	12.54
36	Uccole	*D. geniculata*	3	42.1	13.81
39	Grotta Bella	*D. geniculata*	3	43.4	13.37
40	Pozzo Cambiano	*D. geniculata*	3	42.8	13.66
41	Torri	*D. laetitiae*	3	43.9	11.17
44	San Fili	*D. calabra*	1	39.33	16.09
45	Calcari	*D. calabra*	3	38.38	16.13
45	Calcari	*D. palpata*	1	38.38	16.13
46	Cava de Tirreni	*D. geniculata*	1	40.68	14.69
46	Maiella	*D. geniculata*	2	41.96	14.12

## Data Availability

The original data presented in the study are openly available in Genbank at PX122103–PX122152 and PX124280–PX124329.

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
