# Peer review of "Genetic Diversity and Distribution of Italian Cave Crickets (Dolichopoda): Toward a Better Understanding of Lineage Structure"

_animals, 2025, doi:10.3390/ani15162429_

Round 1

Reviewer 1 Report

Comments and Suggestions for Authors

I thank the authors for submitting their manuscript. The topic is relevant from a taxonomic, evolutionary, and conservation perspective, particularly for faunal groups that remain largely understudied, such as troglobitic orthopterans. The manuscript presents novel and well-analyzed data, supported by up-to-date phylogenetic methods and a noteworthy sampling effort.

That said, several aspects would benefit from clarification, particularly regarding the phylogenetic methods, the interpretation of divergent lineages, and the taxonomic discussion. My comments below are intended to improve the clarity, reproducibility, and scientific impact of the manuscript.

I have only a few minor comments and corrections to suggest in order to improve the clarity of the text.

Comment1:
The salting-out extraction method is referred to as "standard," but it would be preferable to provide more details (e.g., amount of tissue used, duration, temperature, buffer type), or to clearly state whether the referenced method was followed strictly.

Comment2:
Please indicate the expected size of the amplified fragments.

Comment3:
It would be helpful to specify the PCR conditions (reaction volumes, primer concentrations, number of cycles, annealing temperature), even briefly, or to cite a reference if an established protocol was followed without modification.

Comment4:
The name of the sequencing provider is mentioned, but additional information essential for reproducibility is missing, such as the type of sequencing (Sanger?) and the platform used.

Comment5:
Several key parameters are either missing or insufficiently detailed:

  • Quality criteria for selecting sequences from GenBank.
  • Final sequence lengths used (for cox1 and 16S).
  • Parameters used in MAFFT (e.g., number of iterations, gap treatment).
  • Support thresholds used to consider a topology robust.
    These details are critical to ensure the reproducibility of the analyses.

Comment6:
Figure 1 appears slightly blurry, which affects its readability, particularly due to the use of dark colors such as blue, green, and purple. It is recommended to check the sharpness of the image and consider using lighter shades to enhance clarity and distinguishability of elements.

Comment7:
The recognition of divergent lineages within the D. geniculata–laetitiae clade is interesting, but the evolutionary implications deserve further discussion — genetic drift in isolated habitats? role of Quaternary climate? successive colonization events? Exploring these hypotheses could enrich the interpretation.

Comment8:
It is stated that line 192 “neither the original descriptions nor subsequent morphological evaluations provide robust diagnostic characters,” yet no new morphological analysis is undertaken in this study. This could be acknowledged as a limitation and proposed as a priority for future research.

Comment9:
The discovery of sympatry between D. calabra and D. palpata is indeed noteworthy. However, it would benefit from further development: are there pre-/post-zygotic barriers? is interspecific genetic divergence sufficient? does this represent secondary sympatry or long-term coexistence?

Author Response

Comment1:

The salting-out extraction method is referred to as "standard," but it would be preferable to provide more details (e.g., amount of tissue used, duration, temperature, buffer type), or to clearly state whether the referenced method was followed strictly.

Response1: We appreciate the Reviewer’s suggestion. We stated that a standard protocol was used for DNA extraction, following the original source. As this is a widely used and well-established method, we felt that providing further procedural details would not significantly enhance the Materials and Methods section. However, we added the amount of tissue used for each DNA extraction following Reviewer’s suggestion.

Comment2:

Please indicate the expected size of the amplified fragments.

Responses2: We included this information in lines 119-120.

Comment3:

It would be helpful to specify the PCR conditions (reaction volumes, primer concentrations, number of cycles, annealing temperature), even briefly, or to cite a reference if an established protocol was followed without modification.

Responses3: We appreciate this important aspect highlighted by the Reviewer. We now include the PCR conditions citing the appropriate study.

Comment4:

The name of the sequencing provider is mentioned, but additional information essential for reproducibility is missing, such as the type of sequencing (Sanger?) and the platform used.

Responses4: We thank Reviewer for this suggestion. We specify that the used sequencing method was Sanger.

Comment5:

Several key parameters are either missing or insufficiently detailed:

  1. Quality criteria for selecting sequences from GenBank.
  2. Final sequence lengths used (for cox1 and 16S).
  3. Parameters used in MAFFT (e.g., number of iterations, gap treatment).
  4. Support thresholds used to consider a topology robust.
  5. These details are critical to ensure the reproducibility of the analyses.

Responses5:

We really appreciate the comments of Reviewer. And we now covered all the noticed issues:

  1. The criteria and method to download the sequences from GenBank are described in Salvi et al. 2020, and now we better include this citation in the manuscript.
  2. We added this information in lines 119-120.
  3. We now specify that all the rest of settings for the MAFFT alignment were as default.
  4. We added the criteria to read as a robust topology our nodal support. Please, see lines 132-134.

Comment6:

Figure 1 appears slightly blurry, which affects its readability, particularly due to the use of dark colors such as blue, green, and purple. It is recommended to check the sharpness of the image and consider using lighter shades to enhance clarity and distinguishability of elements.

Responses6: we will provide separately the image in high definition.

Comment7:

The recognition of divergent lineages within the D. geniculata–laetitiae clade is interesting, but the evolutionary implications deserve further discussion — genetic drift in isolated habitats? role of Quaternary climate? successive colonization events? Exploring these hypotheses could enrich the interpretation.

Responses7: We thank the Reviewer for this insightful comment. We agree that exploring the evolutionary processes underlying the divergence within the D. geniculata–laetitiae clade — such as genetic drift in isolated habitats, the impact of Quaternary climatic fluctuations — would substantially enrich the interpretation. However, addressing these hypotheses would require more comprehensive data, including expanded sampling across populations, multi-locus genetic datasets, and demographic modelling. Additionally, evaluating the role of Quaternary climate dynamics would necessitate robust divergence time estimates, which are currently not feasible given the available data and lack of reliable molecular calibration points.

Comment8:

It is stated that line 192 “neither the original descriptions nor subsequent morphological evaluations provide robust diagnostic characters,” yet no new morphological analysis is undertaken in this study. This could be acknowledged as a limitation and proposed as a priority for future research.

Responses8: We thank the Reviewer for this valuable comment. We agree that while this study focused on the genetic diversity of Dolichopoda species, morphological variation was not addressed. We recognize that incorporating morphological analyses would provide important complementary insights in future research. As suggested, we have now acknowledged this point in the Discussion section.

Comment9:

The discovery of sympatry between D. calabra and D. palpata is indeed noteworthy. However, it would benefit from further development: are there pre-/post-zygotic barriers? is interspecific genetic divergence sufficient? does this represent secondary sympatry or long-term coexistence?

Responses9: We thank the Reviewer for this insightful comment. At present, it is not possible to determine whether pre- or post-zygotic reproductive barriers exist between D. calabra and D. palpata. However, we fully agree that this is a highly intriguing aspect deserving further investigation, and we have now included this point in the Discussion section.

Reviewer 2 Report

Comments and Suggestions for Authors

Thanks for providing me opportunity to review the paper titled “Genetic diversity and distribution of Italian cave crickets (Dolichopoda): toward a better understanding of lineage structure”. 

Although this manuscript is good but if my suggestions will be incorporated may improve it. Kindly ignore the comments if not relevant

Introduction:

The authors must add in the introduction what ecological roles do cave crickets play within cave ecosystems. Why is mitochondrial DNA (such as cox1 and 16S) particularly useful for studying genetic diversity in cave-dwelling species. How does limited light and food availability in cave environments influence the evolution of cave-adapted insects like Dolichopoda?

Methodology

How robust is the use of cox1 and 16S markers in delineating species-level distinctions within Dolichopoda, especially considering potential incomplete lineage sorting.  About the sampling efforts across the 18 populations sufficient to capture the full spectrum of genetic variation, including potential cryptic diversity. What measures were taken to validate the phylogenetic inferences. Were alternative methods or corroborative morphological data used.

Results and discussion:

To what extent do the sampled cave sites in the northern Apennines represent the broader ecological niches of Dolichopoda species. How were spatial biases or limitations in cave accessibility addressed during sampling. Could environmental variables like microclimate or cave connectivity affect distributional boundaries, was this factored into interpretation. How significant is sympatry between D. calabra and D. palpata at site 45 for understanding speciation or niche overlap in southern Italian habitats. What biogeographic factors might have facilitated the range expansions of D. laetitiae, D. geniculata, and D. calabra.  How can these results inform future molecular sampling strategies to better capture species diversity and lineage differentiation.

Author Response

Comments 1:

Introduction:

The authors must add in the introduction what ecological roles do cave crickets play within cave ecosystems. Why is mitochondrial DNA (such as cox1 and 16S) particularly useful for studying genetic diversity in cave-dwelling species. How does limited light and food availability in cave environments influence the evolution of cave-adapted insects like Dolichopoda?

Response 1: We thank Reviewer 1 for his suggestion, and we improved the Introduction..

Dolichopoda cave crickets are detritivores and scavengers that play a key ecological role in cave ecosystems. By feeding on organic matter, they act as an essential link in the cave food web, transferring energy from detrital resources to higher trophic levels. We added this information in lines 32-36.

Mitochondrial genes such as cox1 and 16S are particularly useful for studying genetic diversity in cave-dwelling species for two main reasons: (i) they evolve relatively rapidly, providing high resolution for detecting phylogenetic relationships and intraspecific diversity; and (ii) they are widely used in previous studies, allowing for direct comparisons and integration with existing data on the Italian cave fauna. We added this information in lines 63-69.

Comments 2:

Methodology

How robust is the use of cox1 and 16S markers in delineating species-level distinctions within Dolichopoda, especially considering potential incomplete lineage sorting.  About the sampling efforts across the 18 populations sufficient to capture the full spectrum of genetic variation, including potential cryptic diversity. What measures were taken to validate the phylogenetic inferences. Were alternative methods or corroborative morphological data used.

Response 2:

Mitochondrial markers such as cox1 and 16S are widely recognized as effective for delineating species-level relationships in invertebrates, including Dolichopoda, due to their relatively rapid rates of evolution and strong phylogenetic signal. The fast evolving rate of these markers is known to reduce the impact of incomplete lineage sorting in phylogenetic  inferences at both intra- and interspecific levels.

Regarding sampling, our dataset—combined with previous studies—covers a broad geographic range across the Italian peninsula, encompassing 18 populations that represent the main lineages known to date. We therefore believe our sampling effort sufficiently captures the existing genetic diversity, including potential cryptic lineages.

Phylogenetic inferences were validated using standard and widely accepted methods, including maximum likelihood and Neighbour Joining, with support values assessed through bootstrapping. Our results are congruent with prior phylogenetic studies of the genus, reinforcing their robustness.

Although morphological data were not included in this study, our primary aim was to investigate genetic diversity and phylogenetic relationships rather than to conduct a formal taxonomic revision. We acknowledge that integrating morphological data could be valuable in future work focused on species delimitation.

Comments 3:

Results and discussion:

To what extent do the sampled cave sites in the northern Apennines represent the broader ecological niches of Dolichopoda species. How were spatial biases or limitations in cave accessibility addressed during sampling. Could environmental variables like microclimate or cave connectivity affect distributional boundaries, was this factored into interpretation. How significant is sympatry between D. calabra and D. palpata at site 45 for understanding speciation or niche overlap in southern Italian habitats. What biogeographic factors might have facilitated the range expansions of D. laetitiae, D. geniculata, and D. calabra.  How can these results inform future molecular sampling strategies to better capture species diversity and lineage differentiation.

Response 3: We thank the reviewer for these insightful comments. We acknowledge that incorporating ecological niche data—such as microclimatic conditions, cave connectivity, and environmental variables—could offer valuable context for understanding the biogeographic history and distributional patterns of Dolichopoda species. However, our study was primarily focused on assessing intra- and interspecific genetic diversity and phylogenetic relationships, and did not explicitly include ecological niche modeling or environmental variable analysis.

Reviewer 3 Report

Comments and Suggestions for Authors

The manuscript by Garzia et al. entitled “Genetic diversity and distribution of Italian cave crickets (Dolichopoda): toward a better understanding of lineage structure” sequences two mitochondrial markers in 50 individuals sampled in 18 populations of cave crickets (Dolichopoda). They find wider geographic distributions for some of the species than previously described and, in particular, find evidence of sympatry for two species that were thought to be allopatric. The phylogenetic analysis of the concatenated sequences of cox1 and 16S using additional sequences from GenBank revealed some undescribed lineages. Results confirm the complex nature of the clade Dolichopoda geniculata–laetitiae and show that the sympatric species also have evolved new lineages in its area of contact.

Comments to the authors:

In general, I found some information missing, for example: what is the length of the sequences analyzed? What option was used to reconstruct the phylogenetic trees, complete deletion, pair deletion...? I also think that it should be interesting to see the table of polymorphic sites where the variants of the new linages were indicated.

In Material and Methods (L.69-70) authors say: “Samples were collected, and one leg from each specimen was removed and preserved in 95% ethanol for subsequent genetic analysis.” From my ignorance about these animals, I wonder what happened to the rest of the animal. Was it released and can it continue to live without a leg or was it reserved for further studies?

Table 1 should be improved in different ways:

  • Tables should have a header with a title and may have a footer with additional information.
  • Perhaps, it should be a good idea to insert a new line to divide the cave ID 45 into two populations, one for each species ( calabra and D. palpata).
  • Also, if ordered by latitude instead of by ID, the information about the species distribution is more ease to visualize.
  • Whereas the main text says that 50 specimens were studied, the table only shows the origin of 47 of them.

The results paragraph should be rewritten to be clearer and avoid confusion. I think it would be a good idea to split the single paragraph into two. The first paragraph referring to the distribution ranges found and the second paragraph referring to the phylogenetic analyses. This would also mean that Figures 1 and 2 should be swapped.

L 130 -131. “…our newly generated sequences to six previously recognized Italian species: D. laetitiae, D. geniculata, D. bormansi, D. schiavazzii, D. calabra, and D. palpata (Figure 1A).” Figure 1A is so schematic that it does not show what the new sequences are nor what the six species referred to in the text are. Perhaps the text should refer to Figure 1B… On the other hand, in the figure legend it is necessary to explain what the frame of red dashed lines is. In addition, colors in Figure 1B should be explained. For example, why are several colors for the same species (D. geniculate)?

L 136-146. I had a hard time understanding what the "site" was referring to. At first, I thought it was referring to the nucleotide positions with variants and only after greatly enlarging the figure did I realize that it was referring to the Cave ID. Perhaps it would be better to change “site” to “locality” or “population”. For example: “Within this clade, our data identified two previously unrecognized evolutionary lineages: one found in sites 11, 12, 14, 39, and 40 (uBS = 100), and another exclusive to the site 46 (uBS = 100).”could be: Within this clade, our data identified two previously unrecognized evolutionary lineages: one found in localities 11, 12, 14, 39, and 40 (uBS = 100), and another exclusive to the locality 46 (uBS = 100).

L137-140. "Additionally, new lineages were recovered within the southern species D. calabra and D. palpata. In particular, sites 44 and 45 hosted a novel lineage of D. calabra (uBS = 99), while the site 45 also harbored a distinct lineage of D. palpata (uBS = 82), indicating an area of sympatry between these two taxa." It should be better explaining why this observation indicates sympatry between these two taxa. What makes them sympatric is not the fact of finding new lineages but the fact of finding the two species coexisting. This coexistence may lead to them becoming more differentiated to avoid competition and could be the cause of observing these new lineages.

L.208. Authors claim that D. calabra and D. palpate are found in syntopy, but they do not explain any observation that conduct to affirm that. These two species are found in the same area and then they are sympatric species. However, to be syntopic they should use the same resources. Is there any evidence of that? Maybe it could be deeper discussed.  

Minor comments and typos:

L.42. Use italics for the species name “D. apollinea

Figure 2. Change “Distribution of the Italian species” to “Distribution of the Italian and Corsican species”

L.85. Change “Heterozygous positions for the nuclear coding gene” to “Heterozygous positions for the mitochondrial coding gene”. Both markers (cox1 and 16S) analyzed are mitochondrial as are indicated in other parts of the text. Thus, in principle, there is no heterozygotes, or did you find any case of heteroplasmy?

  1. 88-89. Remember providing the accession numbers.

L.108. What does CIT mean?

L.110. What does “Then, concatenated sequence alignments” refer to? The cox1 and 16s sequences? Please specify.

Table S1. There are two individuals from a Spanish population whose locality is not correctly named. “Serradel, Llerida” should be “Serradell, Lleida” (in Catalan) or “Serradell, Lérida” (in Spanish).

Author Response

Comments 1:

In general, I found some information missing, for example: what is the length of the sequences analyzed? What option was used to reconstruct the phylogenetic trees, complete deletion, pair deletion...? I also think that it should be interesting to see the table of polymorphic sites where the variants of the new linages were indicated.

 Response 1: We thank Reviewer 2 for the comments. We added the lengths of the 16S and cox1 fragments in line 118-119. We also added the information on the complete/pairwise deletion options for the NJ construction in line 114.

Comments 2:

In Material and Methods (L.69-70) authors say: “Samples were collected, and one leg from each specimen was removed and preserved in 95% ethanol for subsequent genetic analysis.” From my ignorance about these animals, I wonder what happened to the rest of the animal. Was it released and can it continue to live without a leg or was it reserved for further studies?

 Response 2: Entire specimens were collected and preserved in pure ethanol for genetic analyses, with the remaining body retained for future morphological assessment and as voucher specimens linked to the genetic data.

Comments 3:

Table 1 should be improved in different ways:

  1. Tables should have a header with a title and may have a footer with additional information.
  2. Perhaps, it should be a good idea to insert a new line to divide the cave ID 45 into two populations, one for each species ( calabra and D. palpata).
  3. Also, if ordered by latitude instead of by ID, the information about the species distribution is more ease to visualize.
  4. Whereas the main text says that 50 specimens were studied, the table only shows the origin of 47 of them.

 Response 3: We thank Reviewer 2 for the valuable suggestions regarding Table 1. Please find our responses to the specific points below:

  1. We have revised the table by adding a clear header containing the title and moved the main description to the top, as suggested. Additional relevant information remains included in the caption.
  2. We have added a new row to separately list the two species (D. calabra and D. palpata) recorded at cave site 45, to clarify that they represent distinct populations.
  3. While we acknowledge that ordering the table by latitude may help visualize species distributions, we have chosen to maintain the current order by cave ID. This structure aligns more directly with the map in the main text, facilitating cross-referencing between geographic locations and genetic data. We believe this organization better serves readers consulting both the table and figures.
  4. Thank you for pointing out the discrepancy in specimen numbers. We identified the issue and have corrected Table 1 to accurately reflect the total of 45 specimens included in the study.

Comments 4:

The results paragraph should be rewritten to be clearer and avoid confusion. I think it would be a good idea to split the single paragraph into two. The first paragraph referring to the distribution ranges found and the second paragraph referring to the phylogenetic analyses. This would also mean that Figures 1 and 2 should be swapped.

Response 4: We thank the reviewer for the helpful suggestion to split the Results section into two separate paragraphs—one focusing on distribution ranges and the other on phylogenetic analyses. However, in our study design, phylogenetic inference represents the essential first step for identifying individuals to known species or distinct genetic lineages. This classification is a prerequisite for interpreting subsequent biogeographic patterns, such as range extensions, sympatry, or the discovery of novel lineages. Given the logical flow of our analysis and the strong interdependence between phylogenetic resolution and geographic interpretation, we feel that keeping the current structure—with phylogenetic results presented first—is the most coherent and informative approach.

Comments 5:

L 130 -131. “…our newly generated sequences to six previously recognized Italian species: D. laetitiae, D. geniculata, D. bormansi, D. schiavazzii, D. calabra, and D. palpata (Figure 1A).” Figure 1A is so schematic that it does not show what the new sequences are nor what the six species referred to in the text are. Perhaps the text should refer to Figure 1B… On the other hand, in the figure legend it is necessary to explain what the frame of red dashed lines is. In addition, colors in Figure 1B should be explained. For example, why are several colors for the same species (D. geniculate)?

Response 5: We thank Reviewer 2 for pointing this out. We now referred to both Figure 1A and Figure 1B and we also explained that the dashed red rectangle refers to the subtree of Figure 1B.

Comments 6:

L 136-146. I had a hard time understanding what the "site" was referring to. At first, I thought it was referring to the nucleotide positions with variants and only after greatly enlarging the figure did I realize that it was referring to the Cave ID. Perhaps it would be better to change “site” to “locality” or “population”. For example: “Within this clade, our data identified two previously unrecognized evolutionary lineages: one found in sites 11, 12, 14, 39, and 40 (uBS = 100), and another exclusive to the site 46 (uBS = 100).”could be: Within this clade, our data identified two previously unrecognized evolutionary lineages: one found in localities 11, 12, 14, 39, and 40 (uBS = 100), and another exclusive to the locality 46 (uBS = 100).

Response 6: We thank Reviewer 2 for pointing this out. We agree with Reviewer 2 and we change “site/sites” terms with “locality/localities”. We applied this change across the manuscript.

Comments 7:

L137-140. "Additionally, new lineages were recovered within the southern species D. calabra and D. palpata. In particular, sites 44 and 45 hosted a novel lineage of D. calabra (uBS = 99), while the site 45 also harbored a distinct lineage of D. palpata (uBS = 82), indicating an area of sympatry between these two taxa." It should be better explaining why this observation indicates sympatry between these two taxa. What makes them sympatric is not the fact of finding new lineages but the fact of finding the two species coexisting. This coexistence may lead to them becoming more differentiated to avoid competition and could be the cause of observing these new lineages.

Response 7: Following the reviewer’s comment, we rephrased as follow: "Additionally, new lineages were recovered within the southern species D. calabra and D. palpata. In particular, a novel lineage of D. calabra (uBS = 99) was found in sites 44 and 45 hosted, and a distinct lineage of D. palpata (uBS = 82) was found in the site 45. Therefore in site 45 these two new lineages are syntopic.". Please, see lines 154-157.

Comments 8:

L.208. Authors claim that D. calabra and D. palpate are found in syntopy, but they do not explain any observation that conduct to affirm that. These two species are found in the same area and then they are sympatric species. However, to be syntopic they should use the same resources. Is there any evidence of that? Maybe it could be deeper discussed.  

 Response 8: The rewording of the sentence described in Comment 7 explains that these taxa are found in the same cave, i.e. they are syntopic.

Minor comments and typos:

Comments 9:

L.42. Use italics for the species name “D. apollinea”

Response 9: Corrected. 

Comments 10:

Figure 2. Change “Distribution of the Italian species” to “Distribution of the Italian and Corsican species”

Response 10: Corrected.

Comments 11:

L.85. Change “Heterozygous positions for the nuclear coding gene” to “Heterozygous positions for the mitochondrial coding gene”. Both markers (cox1 and 16S) analyzed are mitochondrial as are indicated in other parts of the text. Thus, in principle, there is no heterozygotes, or did you find any case of heteroplasmy?

Response 11: We highly appreciate that Reviewed 2 notice this sentence. We agree with Reviewed 2 and we deleted the part because we didn’t find any case of heteroplasmy.

Comments 12:

88-89. Remember providing the accession numbers.

Response 12

We have already requested the accession numbers to GB and will provide them as soon as they are available, before publication.

Comments 13:

L.108. What does CIT mean?

Response 13: We thank Reviewer 2 for noticing a missing citation. We added the proper citation.

Comments 14:

L.110. What does “Then, concatenated sequence alignments” refer to? The cox1 and 16s sequences? Please specify.

Response 14: We thank Reviewer 2. Specified.

Comments 15:

Table S1. There are two individuals from a Spanish population whose locality is not correctly named. “Serradel, Llerida” should be “Serradell, Lleida” (in Catalan) or “Serradell, Lérida” (in Spanish).

Response 15: We thank Reviewer 2 for noticing the small mistake. Modified.

Reviewer 4 Report

Comments and Suggestions for Authors

Genetic diversity and distribution of Italian cave crickets (Dolichopoda): toward a better understanding of lineage structure

L. 15. Include the names of the modeling programs that were used in this work.

L. 15. Include objective of the project.

L. 22. indicate reason why they are considered model organisms in the study of biogeographic patterns.

L. 25. State species number in the subfamily Troglophilinae.

L. 26-32. This line is very long, use a period to separate the ideas and make reading more comfortable.

L. 26-32. Separate the statement in two.

L.42. There is a scientific name that is not in italics: "D. apollinea"

L. 44. Place the meaning of the word "Allopatric."

L. 55-57. I suggest mentioning which vertebrates feed on Dolichopoda

L. 68. Add the average annual temperature and the average annual precipitation of the collection sites.

L. 73. Remove the mention from Figure 2, since it does not need to be placed in this description.

L. 124. The phylogenetic map you present has sections that are blurred, this limits the understanding of the map, corrects this error.

L. 124. Some places on the phylogenetic map make it difficult to read, place dimmer colors or change the color of the letter.

L. 188. What kind of barriers do you mean by "topographic barriers"

L. 216-219. This statement would be better in conclusion.

L. 232-310. I suggest checking that the genders and scientific names in the references are in italics.

L. 294-295. I suggest adding the doi and checking the other references, as some don't have it.

Comments on the Quality of English Language

Manuscript must be reviewed in writing style since it is not clear. There are several grammar mistakes.

Author Response

Comments 1: Genetic diversity and distribution of Italian cave crickets (Dolichopoda): toward a better understanding of lineage structure

Response 1: OK

Comments 2: L. 15. Include the names of the modeling programs that were used in this work.

Response 2:  We have not used any modeling program in this study

Comments 3: L. 15. Include objective of the project.

Response 3: we stated ‘to investigate their distribution and genetic diversity’

Comments 4: L. 22. indicate reason why they are considered model organisms in the study of biogeographic patterns.

Response 4: we rephrase this lines and added this information: ‘Dolichopoda are considered model organisms for biogeography because their evolutionary patterns are strongly shaped by vicariant events making them ideal for studying historical biogeographic processes’.

Comments 5: L. 25. State species number in the subfamily Troglophilinae.

Response 5: As Troglophilinae is a distinct subfamily from Dolichopodinae—to which Dolichopoda belongs—we consider the number of species in Troglophilinae outside the scope of this study and not directly relevant to our objectives. For clarity and focus, we have limited our discussion to Dolichopoda and its subfamily.

Comments 6: L. 26-32. This line is very long, use a period to separate the ideas and make reading more comfortable.

Response 6: thank you for this comment. Corrected.

Comments 7: L. 26-32. Separate the statement in two.

Response 7: corrected.

Comments 8: L.42. There is a scientific name that is not in italics: "D. apollinea"

Response 8: corrected.

Comments 9: L. 44. Place the meaning of the word "Allopatric."

Response 9: we believe any zoologist know the meaning of allopatric.

Comments 10: L. 55-57. I suggest mentioning which vertebrates feed on Dolichopoda

Response 10: The current sentence reads: “Invertebrates are known to be useful predictors of conservation priorities which can also benefit vertebrate conservation.” As this statement is intended to emphasize the broader conservation value of invertebrates, we feel that adding specific information on Dolichopoda predators would shift the focus and interrupt the flow of the argument. For this reason, we prefer to keep the sentence as is.

Comments 11: L. 68. Add the average annual temperature and the average annual precipitation of the collection sites.

Response 11: We appreciate the suggestion; however, since our study focuses on phylogenetic relationships based on mitochondrial markers, we do not consider climatic variables such as average annual temperature and precipitation to be directly relevant to our objectives or analyses. 

Comments 12: L. 73. Remove the mention from Figure 2, since it does not need to be placed in this description.

Response 12: We respectfully suggest retaining the reference to Figure 2, as it helps readers link the cave IDs listed in the table with their geographic locations on the map. This cross-reference enhances clarity and facilitates spatial interpretation of the sampling sites.

Comments 13: L. 124. The phylogenetic map you present has sections that are blurred, this limits the understanding of the map, corrects this error.

Response 13: corrected. In the high resolution image this problem is solved.

Comments 14: L. 124. Some places on the phylogenetic map make it difficult to read, place dimmer colors or change the color of the letter.

Response 14: corrected. In the high resolution image this problem is solved.

Comments 15: L. 188. What kind of barriers do you mean by "topographic barriers"

Response 15: topographic barriers include mountain ridges, deep valleys, and isolated karst systems that characterize the central Apennine region. We added this information in the text.

Comments 16: L. 216-219. This statement would be better in conclusion.

Response 16: we now added a conclusion section.

Comments 17: L. 232-310. I suggest checking that the genders and scientific names in the references are in italics.

Response 17: checked.

Comments 18: L. 294-295. I suggest adding the doi and checking the other references, as some don't have it.

Response 18: checked.

Round 2

Reviewer 4 Report

Comments and Suggestions for Authors

Several of my suggestions were no performed and they were not justified scientifically. So, I have no more comments.

Comments on the Quality of English Language

English need to be revised by a native speaker or translator since thee are several grammatical mistakes.